# Controlled-Release Hydrogel Microspheres to Deliver Multipotent Stem Cells for Treatment of Knee Osteoarthritis

**DOI:** 10.3390/bioengineering10111315

**Published:** 2023-11-15

**Authors:** Megan Hamilton, Jinxi Wang, Prajnaparamita Dhar, Lisa Stehno-Bittel

**Affiliations:** 1Bioengineering Program, School of Engineering, University of Kansas, Lawrence, KS 66045, USA; prajnadhar@ku.edu; 2Likarda, Kansas City, MO 64137, USA; lbittel@likarda.com; 3Department of Orthopedic Surgery and Sport Medicine, School of Medicine, University of Kansas Medical Center, Kansas City, KS 66160, USA; jwang@kumc.edu

**Keywords:** hydrogel, microsphere, cell deliver, MSC therapy, osteoarthritis

## Abstract

Osteoarthritis (OA) is the most common form of joint disease affecting articular cartilage and peri-articular tissues. Traditional treatments are insufficient, as they are aimed at mitigating symptoms. Multipotent Stromal Cell (MSC) therapy has been proposed as a treatment capable of both preventing cartilage destruction and treating symptoms. While many studies have investigated MSCs for treating OA, therapeutic success is often inconsistent due to low MSC viability and retention in the joint. To address this, biomaterial-assisted delivery is of interest, particularly hydrogel microspheres, which can be easily injected into the joint. Microspheres composed of hyaluronic acid (HA) were created as MSC delivery vehicles. Microrheology measurements indicated that the microspheres had structural integrity alongside sufficient permeability. Additionally, encapsulated MSC viability was found to be above 70% over one week in culture. Gene expression analysis of MSC-identifying markers showed no change in CD29 levels, increased expression of CD44, and decreased expression of CD90 after one week of encapsulation. Analysis of chondrogenic markers showed increased expressions of aggrecan (ACAN) and SRY-box transcription factor 9 (SOX9), and decreased expression of osteogenic markers, runt-related transcription factor 2 (RUNX2), and alkaline phosphatase (ALPL). In vivo analysis revealed that HA microspheres remained in the joint for up to 6 weeks. Rats that had undergone destabilization of the medial meniscus and had overt OA were treated with empty HA microspheres, MSC-laden microspheres, MSCs alone, or a control vehicle. Pain measurements taken before and after the treatment illustrated temporarily decreased pain in groups treated with encapsulated cells. Finally, the histopathological scoring of each group illustrated significantly less OA damage in those treated with encapsulated cells compared to controls. Overall, these studies demonstrate the potential of using HA-based hydrogel microspheres to enhance the therapeutic efficacy of MSCs in treating OA.

## 1. Introduction

Osteoarthritis (OA) is a chronic, progressive degenerative joint disease clinically characterized by joint pain, stiffness, and deformity with radiographic evidence of joint space narrowing due to a loss of articular cartilage (AC), osteophyte formation, and subchondral bone sclerosis [1,2]. For the United States healthcare system, OA is a heavy burden, with an annual cost greater than $45 billion [3]. The current standard of care is aimed at pain prevention and management through the use of steroids, non-steroidal anti-inflammatory drugs, and viscosupplements [4,5]. However, these treatments are not disease-modifying, and so the progression of OA typically leads to aggressive cartilage degeneration, and ultimately, patients require a total joint replacement [6]. Not surprisingly, in search of a treatment capable of both treating symptoms and modifying the disease progression, cell-based regenerative strategies are of interest [7].

Multipotent stromal cell, also known as mesenchymal stem cell (MSC) therapy, has been studied for decades as a potential treatment for OA, with only one treatment approved globally, Cartistem, by Medipost [8]. The first successful MSC transplant into caprine osteoarthritic joints was in 2003; Murphy et al. illustrated meniscal repair and chondroprotection following MSC treatment [9,10]. Since then, a considerable amount of research has gone towards advancing the clinical use of stem cells to treat OA. MSCs are an attractive source for this application due to their multipotency, immunomodulatory characteristics, and ability to be extracted from a variety of body tissues including bone marrow, adipose tissue, umbilical cord blood, and Wharton’s Jelly [11]. While the precise therapeutic mechanism is unclear, it is thought to be a mix of paracrine-mediated mechanisms, immunomodulation, and to a lesser extent, the differentiation of transplanted stem cells into functional chondrocytes [9,12,13]. Pre-clinical and clinical studies have shown minor success, but oftentimes long-term efficacy is not achieved or results are not reproducible [14,15].

A major hindrance to the successful clinical translation of MSC therapy for OA is the lack of control over viability and retention in target regions following administration [16]. The most common route of administration for MSC therapy is a direct injection into the intra-articular (IA) space [14]. However, several groups have reported significant cell death and minimal cell engraftment using this method [15]. Indeed, it has been reported that 24 h after injection, only 2% of MSCs remain in the joint space of immune-competent rats [17]. Cell loss is posited to be a result of two main factors: (1) when anchorage-dependent cells, such as MSCs, are suspended in solution, they are susceptible to a specialized form of apoptosis, termed anoikis, which is caused by the loss of cell-extracellular matrix (ECM) attachment regions [18,19,20]; and (2) if cells do survive delivery, they often leak out of the joint during movement and weight-bearing [14].

To account for delivery-mediated cell death, several studies utilize serial injections or supratherapeutic doses [14,21,22,23,24]. However, these approaches pose a massive burden on stem cell therapy costs and production and introduce safety concerns [14,25]. A more attractive option is to deploy a biomaterial-based delivery system [26]. Hydrogels are an ideal biomaterial for the assisted delivery of MSCs. Hydrogels are crosslinked networks of hydrophilic polymer chains capable of mimicking ECMs and have been extensively studied for tissue engineering, regenerative medicine, and drug delivery [27,28]. Hydrogels can directly address the issues of transplanted cell viability and retention by providing the attachment sites necessary for survival, resulting in higher cell viability post-delivery. Additionally, depending on the degradation rate of the hydrogel, they offer prolonged retention time in the target region [18,19,29].

Hyaluronic acid (HA)-based hydrogels are of particular interest for this application due to their biocompatibility, biodegradability, and tunability [30]. The degradability of HA is an especially important material property for MSC delivery, as hydrogel degradation has been shown to facilitate the localized release of cells [29,31,32]. HA can be degraded by hyaluronidase, a naturally occurring enzyme in the human joint [33]. The tunability of hydrogels allows them to be synthesized in a variety of shapes and sizes. Bulk, macroscopic hydrogels are often studied for cell encapsulation and delivery. However, bulk hydrogels limit clinical translation because the administration is invasive and cumbersome [34]. Injectable hydrogel microspheres are ideal because they can be injected directly into the joint [35]. Further, because encapsulated stem cell viability is partially dependent on the diffusion of nutrients into the hydrogel network, microspheres can facilitate greater viability by presenting a lower diffusion barrier compared to bulk hydrogels [36,37].

Previous work by our group has shown that delivery of islet cells encapsulated in poly(ethylene) glycol-based hydrogel microspheres into a diabetic rodent model facilitated healthy blood glucose levels compared to an unencapsulated cell transplant control group [38]. This work demonstrates the value of hydrogel microsphere-mediated cell delivery for long-term applications where immunoprotection is essential for a successful transplant [34]. Additional earlier works have expanded the hydrogel microsphere platform by developing microspheres to be degradable for applications in which slow release of the encapsulated cells is ideal [36]. Thus, the objective of the studies described herein is to expand on this previous work by evaluating the efficacy of such degradable microspheres in treating OA. It is important to note that in the context of stem cell therapy for OA, the MSCs are theorized to function therapeutically through the secretion of soluble factors and direct interaction with host cells. This is in slight contrast to tissue engineering efforts, which often employ hydrogel scaffolds to encapsulate MSCs and guide their differentiation into functional chondrocytes. Thus, the studies described herein do not attempt to guide differentiation or achieve new cartilage tissue formation but merely aim to deliver MSCs such that their secretions can be retained in the local joint space.

Certainly, there have been many previous efforts to achieve a hydrogel microsphere-based delivery system for stem cell therapy in OA. Using alginate-based hydrogel microspheres, Sahu et al. illustrated the potential of MSC-laden hydrogel microbeads to promote osteoarthritic repair [37]. In their study, ex vivo OA cartilage tissue was co-cultured with MSC-laden alginate microspheres, and it was found that the microspheres enhanced proteoglycan synthesis and ECM remodeling, induced cell proliferation, and modulated levels of inflammatory cytokines [37]. While these ex vivo data were promising, the use of alginate for in vivo delivery is not ideal because humans do not possess endogenous enzymes to degrade alginate. Further, the therapeutic mechanism of MSCs relies partially on transplanted cell-to-host cell contact [39], which would be difficult to achieve using a non-degradable material. To that end, this study investigated the use of degradable HA-based hydrogel microspheres for in vivo delivery of MSCs into an osteoarthritic joint. Degradable microspheres were synthesized using Core-Shell Spherification technology, and then their microrheological properties were analyzed. Encapsulated cell viability and gene expression were assessed. After quantifying the retention time of the hydrogel in a rodent knee, encapsulated cells were delivered into osteoarthritic rodent joints to evaluate their potential for treating such a condition. Afflicted joints were assayed for pain throughout the study, followed by histological analysis to assess the extent of OA-induced damage to the articular cartilage and surrounding tissues.

## 2. Materials and Methods

### 2.1. Production of Hydrogel Spheres

#### 2.1.1. Synthesis of AHA

Acrylated HA (HA) was synthesized by reacting HA (MW 200 kDa, Lifecore Biomedical, Chaska, MN, USA) with acryloyl chloride and glycidol (TCI) in the presence of triethylamine and N-dimethylformamide (Sigma Aldrich, Saint Louis, MO, USA) for 5 days [40]. Next, the reaction mixture was dialyzed against deionized (DI) water for 120 h, frozen at −80 °C for 24 h, and then lyophilized. The degree of acrylation was determined to be 53–72% using ^1^H NMR (Avance AV-III 500, Bruker, Billerica, MA, USA) by calculating the ratio of the relative peak area of acrylate protons to acetamide methyl protons.

#### 2.1.2. Fabrication of AHA Microspheres

HA-based hydrogel microspheres were manufactured using Core Shell Spherification as described previously, but changes to the polymer concentrations were made in order to obtain microspheres with rheological properties similar to that of healthy synovial fluid in order to reduce pain associated with an injection into the space and provide an appropriate environment for the cells [34,36,41]. Briefly, 5% AHA and 2% 1.0 kDA PEGDA (Laysan Bio, Arab, AL, USA) (*w*/*w*) were prepared in a custom buffer composed of 100 mM calcium chloride, 15 mM HEPES, and 0.05% (*w*/*v*) Irgacure 2959. The solution was filtered through a 0.8-micron filter, then a 0.22-micron filter, and then a semi-micro calibrated glass capillary viscometer was used to determine the viscosity at room temperature. For microsphere fabrication, a Buchi 395-Pro Encapsulator (Buchi Corporation, Newcastle, DE, USA) equipped with an air jet nozzle within a concentric air nozzle was used to extrude the precursor solution into a stirred bath of 0.15% (*w*/*v*) sodium alginate (Protanal LF 10/60, FMC Corp, Philadelphia, PA, USA) with 300 mM mannitol, 0.1% Tween 20, 0.05% (*w*/*v*) Irgacure 2959, and 15 mM HEPES buffer. The alginate bath was degassed via sonication to remove excess oxygen, which is known to inhibit photo-cross-linking [42]. The HA droplets formed core-shell constructs that were exposed to long-wave UV light (PortaRay 400, Uvitron International, West Springfield, MA, USA) to initiate free radical chain growth photopolymerization of the HA (Figure 1). Throughout extrusion, irradiation was applied continuously and for 5-min after extrusion was completed to ensure ample cross-linking in all microspheres. Microspheres were next rinsed in a 25 mM citrate buffer in 1X DPBS for 300 s to remove the alginate. After the dissolution of the alginate shell, microspheres were collected using a steel mesh screen in DPBS. Another citrate rinse of 50 mM sodium citrate was carried out for 300 s to ensure that no alginate remained on the microspheres.

### 2.2. Assessment of Hydrogel Sphere Physical Properties

#### 2.2.1. Assessment of Average Hydrogel Sphere Diameter

Images of 50 individual microspheres from three independent batches were captured using a Cytation 5 Imaging Multi-Mode Reader (Biotek Instruments, Inc., Winooski, VT, USA) [34,36]. The diameter of each individual microsphere was measured using Photoshop 2022 and used to determine the average diameter and size distribution for each of the hydrogel microsphere types.

#### 2.2.2. Assessment of Hydrogel Sphere Swelling Ratio

Hydrogel microspheres were characterized by quantification of the swelling ratio [34,36]. The swelling ratio, Q, is defined as the ratio of the hydrated mass to the dry mass. A minimum of 0.500 g of hydrated microspheres were equilibrated in deionized water, transferred to a pre-weighed watch glass, and then excess moisture was removed. The microspheres were then stored at 60 °C for 24 h and reweighed to obtain the dry mass (N = 3).

#### 2.2.3. Microrheological Analysis of Hydrogel Microspheres

To analyze the rheological properties of the HA-based microspheres, multiple particle tracking (MPT) microrheology was used. MPT is a commonly used technique for measuring the rheology of materials where traditional techniques cannot be used [43]. MPT relies on the inherent thermodynamic fluctuations of small particles embedded within a material [44]. In this study, 0.175 ± 0.005 µm orange fluorescent tracer probes obtained from the PS-Speck Microscope Point Source Kit (Thermo Fisher Scientific, Waltham, MA, USA) were encapsulated in the HA-based microspheres using the fabrication method described above. Upon encapsulation of the tracer probes, their movement was visualized using an upright microscope (Leica DM 2500M, Wetzlar, Germany) at 50× magnification. A fluorescent light source was used to capture the emission and tracking of embedded tracer probes by a CCD Andor Luca video camera with a shutter speed of 0.05 s. After tracking, a custom MATLAB program was used to obtain particle trajectories from which the mean squared displacement (MSD), storage modulus (G′), and loss modulus (G″) of the hydrogel were calculated according to the Generalized Stokes-Einstein Relation [44].

### 2.3. Effect of Encapsulation on Cell Viability and Identity

#### 2.3.1. Culture of Rat Bone Marrow-Derived Multipotent Stromal Cells

Rat bone marrow-derived multipotent stromal cells (bMSCs) (Cyagen Biosciences Inc.; Santa Clara, CA, USA) were thawed according to the manufacturer’s instructions. After thawing, bMSCs were transferred to a falcon tube containing high glucose Dulbecco’s modified eagle medium (HG-DMEM) containing 10% FBS, 1% Glutagro (Gibco, Thermo Fisher Scientific, Waltham, MA, USA), and 1% ANTI-ANTI (Corning, Corning, NY, USA). The bMSCs were centrifuged, the cell pellet was transferred to fresh bMSC media, then seeded in a T-25 flask, and cultured at 37 °C and 5% CO_2_.

#### 2.3.2. Encapsulation of bMSCs

The hydrogel precursor solution was prepared as described above and combined with a slurry of bMSCs to reach a final cell seeding density of 80 million cells per mL of the precursor solution. This cell seeding density was chosen to account for the post-extrusion swelling noted in the hydrogel microspheres, which resulted in 4 mL of hydrated microspheres from 1 mL of precursor. Thus, the seeding density was chosen to achieve a final cell density of 2 million cells per 100 µL of hydrogel microspheres, which is ideal for injection into the small volume of a rodent IA space. Cell-laden microsphere production was achieved using the methods described above. Further, alginate shells were removed as described above, except that 1X Hank’s Balanced Salt Solution was used in place for 1X DPBS. The bMSC-laden HA microspheres were then cultured in the same media as above.

#### 2.3.3. Viability of Cells in AHA Microspheres

The bMSC viability was analyzed with a live/dead fluorescence assay using calcein AM (Life Technologies, Carlsbad, CA, USA) and ethidium homodimer (Sigma-Aldrich, Saint Louis, MO, USA). Calcein exhibits green fluorescence when taken up by living cells and ethidium homodimer is a red fluorescent dye that can only pass through the cell membrane of dead cells to bind strongly to DNA. Encapsulated spheres were incubated with each dye, rinsed with 1X DPBS, and then imaged by fluorescence microscopy. A total of 10 spheres per group were imaged at 469 nm and 531 nm excitation wavelengths for calcein AM and ethidium homodimer, respectively, with the number of green and red pixels measured, indicating the number of live and dead cells using the Cytation 5 Cell Imaging Multi-Mode Plate Reader (Agilent, SantaClara, CA, USA). The background fluorescence of the hydrogel was subtracted when appropriate. Viability was calculated by taking the ratio of green pixels to total cells in each sphere. The results are reported as the average percent viability.

#### 2.3.4. Evaluation of Gene Expression Changes via qPCR on Encapsulated MSCs

One week following encapsulation in HA microspheres, an enzymatic digestion of the microspheres was carried out to isolate the encapsulated bMSCs. Briefly, microspheres were exposed to 5000 U/mL hyaluronidase (Sigma Aldrich, Saint Louis, MO, USA) for 60 min. An unencapsulated population of MSCs underwent the same enzymatic digestion protocol to serve as controls. After collection, cells were lysed using TRI Reagent (R2050-1-200, Zymo Research, Irvine, CA, USA). The Direct-zol RNA MiniPrep Plus Kit (R2072, Zymo Research, Irvine, CA, USA) was used to extract RNA from isolated cells according to the manufacturer’s instructions. A TaqMan probe kit (Thermo Fisher Scientific, Waltham, MA, USA) was used for qPCR analysis of rbMSC identifying markers as indicated by Cyagen and select chondrogenic and osteogenic differentiation markers as shown in Table 1, along with the primer identification number as Thermo Fisher Scientific does not make the sequences publicly available. The housekeeping gene was GUSB, as this has been shown to be a stable reference gene for MSCs, particularly in HA-based hydrogels [45]. This study was carried out on three independent batches and samples were tested in triplicate. Fold change differences in gene expression were calculated using the ΔΔC_t_ method and the results were displayed as the relative gene expression compared to unencapsulated cells.

### 2.4. In Vivo Function of Hydrogel Microspheres

#### 2.4.1. Hydrogel Degradation and In Vivo Retention

To ensure the HA microspheres were susceptible to enzymatic degradation that allowed for encapsulated cell release, bMSC-laden HA hydrogel microspheres were stored in bMSC media supplemented with 50 U/mL hyaluronidase (Sigma Aldrich, Saint Louis, MO, USA) for 7 days to simulate accelerated enzymatic degradation. Representative images were taken on day 0 and day 7 to qualitatively monitor the change in microsphere size as well as to visually confirm the release of encapsulated cells.

In vivo retention was quantified in healthy rodents. All animal studies were carried out under the approved IACUC protocol from the University of Kansas Medical Center (2019-2527) and were performed in compliance with policies on animal use and ethics. Two different approaches were used to quantify the amount of hydrogel in the joint space over time. First, the hydrogel was fluorescently labeled 5-((2-(and-3)-S-(acetylmercapto)succinoyl)amino) (SAMSA) fluorescein (Thermo Fisher Scientific, Waltham, MA, USA), a fluorescent compound functionalized with a thiol group that is capable of covalently binding to acrylate groups on the AHA polymer backbone, by incubating crosslinked HA microspheres with 20 µM SAMSA for 2 h at room temperature. After incubation, microspheres were transferred into an excess of fresh 1X DPBS for 10 min to allow unreacted SAMSA to diffuse out of the microspheres. The fluorescently labeled HA microspheres were injected into the IA space of healthy 9- to 12-week-old male Sprague-Dawley rats. After injection, visualization of the HA microspheres in vivo was attempted using a non-invasive in vivo imaging system (Caliper LifeSciences, Hopkinton, MA, USA). Briefly, rats were anesthetized via Isoflurane inhalation, then fluorescently imaged at excitation and emission wavelengths of 465 nm and 540 nm, respectively.

Secondly, animals received an IA injection of microspheres containing nile blue-marked melamine resin microparticles (Sigma-Aldrich, Saint Louis, MO, USA) for visualization. After 1, 3, and 6 weeks, the animals were euthanized, and their joints were collected for processing. To determine if any hydrogel material was remaining in the joint, the joint was dissected, the capsule lifted, and a surgical spatula was used to remove any materials from the joint. Additionally, 1X DPBS was used to rinse the joint space and flush out the remaining hydrogel components. Recovered materials were deposited into 1X DBPS in a 96-well plate for imaging on a Cytation 5 Plate Reader (Agilent, SantaClara, CA, USA). ImageJ was then used to determine the area of the hydrogel material. The results are displayed as the area of hydrogel material normalized to the area in the initial dose prior to injection.

#### 2.4.2. Induction and Treatment of Osteoarthritis Rat model

Knee osteoarthritis (OA) was induced in 10- to 12-week-old male Sprague Dawley rats by destabilization of the medial meniscus (DMM) following published procedures [46,47]. Rats were anesthetized via Isoflurane inhalation prior to the surgery and the procedure was performed on the knee joints under a dissecting microscope. The knee joint was exposed through a medial parapatellar incision, and the joint capsule was incised. The patella and the patellar tendon were kept intact and carefully protected during the procedure. The medial meniscotibial ligament anchors the medial meniscus to the tibial plateau. After careful exposure of the medial meniscotibial ligament, it was transected with surgical scissors to destabilize the medial meniscus which induces post-traumatic OA. The cruciate ligaments and other ligaments around the knee joint were preserved to maintain the stability and mobility of the joint after surgery. The joint capsule was closed with absorbable sutures and the skin was sutured using non-absorbable sutures. Each joint was randomly assigned a treatment: either transplant media (sham) (N = 6), cells only (N = 8), hydrogel only (N = 8), or encapsulated cells (N = 8). Each dose reached a volume of 100 µL, and both treatments with cells contained approximately 2 × 10^6^ bMSCs. Afflicted joints were treated for 4 weeks following the DMM surgical procedure.

#### 2.4.3. Pain Assessment

To quantify and compare pain between groups, an established joint pain test was used with pressure manually applied to the afflicted joints 5 times, and the number of vocalizations in response was recorded [48]. Each joint was tested before the injury, before treatment, and then at 1, 3, 5, and 6 weeks post-treatment by the same evaluator. Data were reported as the average number of vocalizations normalized to the pre-treatment value.

#### 2.4.4. Histological Analysis

Animals were euthanized at 6 weeks post-transplant and the knee joints were harvested and processed for histological analysis. Histological processing was carried out by Atlantic Bone Screen. Briefly, joints were fixed in 10% neutral buffered formalin for 3 weeks and then stored in a decalcification buffer for 2 weeks. Upon fixation and decalcification, sections were embedded in paraffin blocks, and then 4 μm sections were taken from each joint. For cartilage visualization, sections were stained with Safranin-O and Fast Green. For immunofluorescence, sections were stained with DAPI nuclear localization (Thermo Fisher Scientific, Waltham, MA, USA) with primary antibodies for collagen II (Abcam, Cambridge, UK) or ADAMTS5 (Abcam, Cambridge, UK). Briefly, antigen retrieval was performed in a citrate buffer at pH 6 for 15 h at room temperature. For collagen II, sections were incubated in 5% normal goat serum and 1X DPBS for 1 h at room temperature, then incubated with the primary antibody for 24 h at 4 °C. For ADAMTS-5, sections were incubated in 5% bovine serum albumin for 1 h at room temperature, then in the primary antibody for 1 h at room temperature. Next, the sections were left to incubate in an anti-rabbit secondary antibody (Thermo Fisher Scientific, Waltham, MA, USA) for one hour at room temperature.

The quantification of cellular density at the joint surface was based on the DAPI-stained images with a focus on the articular cartilage surface, where there was less autofluorescence than in the subchondral bone and surrounding tissues. The images were taken with the same exposure time, intensity, and gain, and then uploaded into ImageJ, where the same brightness/contrast settings were applied prior to cell counting to ensure consistency between all images.

Quantification of subchondral bone thickness was carried out as described previously by a blinded technician [49]. Briefly, the subchondral bone thickness was measured from five randomly selected but evenly distributed regions underneath the calcified cartilage on both the medial tibial plateau (MTP) and medial femoral condyle (MFC), as the medial side is most susceptible to osteoarthritic damage following the DMM injury model. In addition, the Osteoarthritis Research Society International (OARSI) semi-quantitative histopathological scoring system was used to assess OA damage to the articular cartilage [50]. This scoring system is well-established and has been used extensively to evaluate OA severity in both mice and rat models [51,52,53,54]. Briefly, the most severely damaged section from each joint was chosen, and the extent of vertical clefts and erosion in articular cartilage was evaluated by blinded reviewers to result in a score between 0–6, where the higher the score, the more severe the OA.

### 2.5. Statistical Analysis

All data are reported as the average ± standard error. A One-Way Analysis of Variance (ANOVA) was used to determine the statistical significance of the viability, OARSI scores, and Pain Scores. A Bonferonni post-hoc *t*-test was used for pairwise comparisons. When normality failed in the Pain Scores, an ANOVA on Ranks was performed. A student’s *t*-test was used to test for significant differences in the qPCR data between C_t_ values for encapsulated versus unencapsulated cells.

## 3. Results

### 3.1. Diameter, Swelling Ratio, and Microrheology of Hydrogel Microspheres

HA-based microspheres synthesized using CSS resulted in diameters ranging between 627 and 1347 microns with a swelling ratio of 108 ± 3. Table 2 summarizes the properties of both the hydrogel precursor and the hydrogel microspheres. Further physical characterization was conducted through microrheological analysis using multiple particle tracking (MPT) microrheology.

The G′ and G″ of the HA-based hydrogel microspheres are shown in Figure 2A. The frequency dependence of both the G′ and G″ indicate the HA microspheres are a viscoelastic liquid and that at frequencies greater than 0.3 Hz, the G′ is greater than the G″. This is an important characteristic for cell encapsulation as too much of a viscous contribution to the material properties can lead to insufficient cell anchorage and cause anoikis [55,56]. Shown in Figure 2B is the mean squared displacement (MSD) as a function of time interval, t, of the HA microspheres used in this study (AHA 1) compared to AHA microspheres described previously [36] (AHA 2), as well as compared to model materials, viscous liquids, and elastic solids [57]. The MSD is a measure of how much a particle deviates from its original position over time and is related to the permeability of the hydrogel network; the higher the MSD, the more permeable the hydrogel [58]. Thus, the results confirm the viscoelasticity of the HA microspheres and ensure that the properties of the formulation used in this study allowed for the diffusion of materials essential to cell health.

### 3.2. Encapsulated Cell Viability

In order to ensure that the encapsulated cells were viable at the time of injection, viability was monitored over one week in culture using live/dead stains. Figure 3A illustrates the typical finding with more live cells (green) 1 day after encapsulation. Analysis of the images showed that viability values increased between 3 and 7 days in culture (Figure 3B).

### 3.3. qPCR Analysis of Encapsulated bMSCs

After 1 week in culture, qPCR analysis was performed to determine the impact of encapsulation on bMSC biomarkers. Additionally, analysis of chondrogenic and osteogenic differentiation markers was carried out as some groups have reported changes in gene expression through encapsulation alone [59], and there are concerns regarding the differentiation of cells into undesirable phenotypes [60]. Both the encapsulated and the unencapsulated groups were exposed to the same concentration of hyaluronidase prior to analysis. The biomarkers probed and the C_t_ values for unencapsulated and encapsulated cells are shown in Table 3. Further analysis of the relative gene expression was calculated by normalizing the data from the encapsulated cells to the unencapsulated group using the ΔΔC_t_ method (Figure 3C).

First, the standard biomarkers for MSCs were analyzed for the starting cell bank. The MSC biomarkers are expected to be negative for CD34 and CD45, but positive for CD29, CD90, and CD44 (Table 3). The results confirm that both encapsulated and unencapsulated MSCs did not express CD34 or CD45 but were positive for CD29, CD90, and CD44, establishing their identity as MSCs defined by the supplier (Cyagen Biosciences Inc.; Santa Clara, CA, USA). Subsequently, the relative gene expression comparing the encapsulated cells to the unencapsulated controls showed that there was increased expression of CD44, which is a cell surface receptor for HA, and its upregulation in MSCs exposed to HA has been reported previously [61]. In addition, there were increased expression of chondrogenic differentiation markers ACAN and SOX9, suggesting a partial conversion to a chondrogenic phenotype. A slight downregulation of CD90 expression was noted, which can be associated with a loss of stemness [62]. Simultaneously, 1 week of encapsulation was associated with a downregulation of osteogenic differentiation markers ALPL and RUNX2 compared to the unencapsulated control. The results are summarized in Figure 3C, with the solid bars showing changes in MSC biomarkers, the striped bars showing chondrogenic genes, and the dotted bars showing changes in osteogenic biomarkers in response to encapsulation. The dotted line indicates no change in the gene expression levels compared to the unencapsulated cells.

### 3.4. Hydrogel Retention Time

One of the goals of the encapsulation of MSCs for administration into the knee joint was to hold the cells in the local space while releasing them slowly over time. Thus, one of the motivations for using HA microspheres is HA’s inherent biodegradability in response to a naturally occurring enzyme in the body, hyaluronidase. To ensure that the HA microspheres were indeed susceptible via hyaluronidase and would release encapsulated cells in response to degradation, an in vitro accelerated enzyme-catalyzed degradation test was carried out. Representative images shown in Figure 4 illustrate both the change in microsphere size over 7 days in accelerated degradation conditions and the release of encapsulated cells from the surface of the microspheres.

Thus, after confirmation that enzymatic degradation of HA microspheres resulted in cell release, the next study sought to quantify in vivo hydrogel retention time in the knee joint. The microspheres were fluorescently labeled with SAMSA fluorescein, a fluorescent compound able to bind to the acrylate groups on the HA polymer backbone. The fluorescently labeled HA microspheres were injected into the IA space of healthy 9 to 12 week old male Sprague-Dawley rats. However, there was significant autofluorescence from the rat tissue at the excitation wavelength needed for visualizing the microspheres. Thus, quantification was not possible using this method. Alternatively, HA microspheres containing fluorescent microparticles were manually removed from the rat knee joints at different time points, and the gel amount was measured. Fragments of the hydrogel were found present in the joint up to 6 weeks after implantation into the IA space (Figure 4, final panel).

### 3.5. Pain Assessment following Cell Therapy Treatment

Chronic pain is a key symptom of OA, and pain evaluation is crucial in determining the impact of experimental treatments on disease progression. In this study, a previously described knee compression test was conducted to directly measure pain [48]. There was a concern that the implantation of large hydrogel microspheres could result in increased pain levels in afflicted joints. The results, displayed in Figure 5, were normalized to the pre-treatment score for each group. The before and after injury time points indicated successful disease induction, as the pain scores significantly increased after the surgical injury procedure. Those treated with the transplant vehicle (Sham) showed an increase in pain over time, but the values were not statistically significant compared to the other groups. The hydrogel-only groups showed no change in the pain score throughout the entire 6 weeks after the treatment. Interestingly, both the cells and encapsulated cells showed a decrease in pain after treatment. Notably, only the encapsulated cell group had a statistically significant decline in pain.

### 3.6. Histological Analysis

At the termination of the experiment and 4 weeks after the transplant of the 4 treatments (sham, hydrogel alone, cells alone, and encapsulated cells), the joints were dissected and prepared for sectioning and staining. The brightfield images illustrated the change to the bone surface compared to healthy bone (Figure 6A). Only the healthy (non-injured) and encapsulated cell groups demonstrated a smooth bony surface with appropriate cartilage morphology. Changes in cartilage, including the mineralization of the tissue and new bone development, in the area contribute to further cartilage degradation, as has been previously shown [63].

Others have attempted to correlate the number of chondrocytes present in articular cartilage with the progression of OA [64,65]. In early OA, rapid proliferation and hypertrophic differentiation of chondrocytes are seen [66]. However, chondrocyte apoptosis is also seen in later stages of OA, resulting in hypocellularity [65]. Cell density at the joint surface was calculated using a nuclear stain (Figure 6B); no statistical differences in cell density were noted in the DAPI-stained sections (results not shown).

Collagen II synthesis increases early in the early stages of OA, but at later stages, collagen type II is converted to collagen type I and is mainly found in the subchondral bone tissue [66]. Localization of the collagen II staining showed that all the joints undergoing the injury model had a cellular, punctate staining pattern, which was not noted in the healthy control joints (Figure 6C). While no analysis was conducted on the images, using the same gain settings, the joints receiving encapsulated cells had the highest overall staining intensity for collagen II.

Sections were also stained for the matrix metalloproteinase, ADAMTS5, which is secreted by chondrocytes early in the OA process [67]. No obvious differences were revealed, with the exception of a layer of highly intense cells on the bony surface in the sham-treated group that was not found in any of the other groups (Figure 6D). ADAMTS5 is the main aggrecanase associated with the breakdown of cartilage [67]. The lack of upregulation of ADAMTS5 from the groups may be due to the late progression of OA in these animals in which the cartilage was already destroyed prior to the time of analysis.

Upon staining, the subchondral bone thickness was measured in both the medial tibial plateau (MTP) and medial femoral condyle (MFC). Changes in the subchondral bone in OA have been noted extensively throughout the literature [49,68]. Often, in late-stage OA, there is a marked increase in subchondral bone volume [69]. The results of the analysis are shown in Figure 7B,C for the MTP and MFC, respectively, where the average thickness of the subchondral bone for each experimental group are shown. In comparison, the average subchondral bone thickness in healthy joints was 78.22 ± 34.37 and 140.38 ± 65.62 µm for the MTP and MFC, respectively, which is 2 to 3 times less than the experimental group values (Figure 7B,C). While treatment with encapsulated cells did result in the lowest subchondral bone thickness in both the MTP and MFC, there were no significant differences between groups in either region.

The OARSI semi-quantitative histopathological scoring system was used to determine the severity of OA [50]. Representative images from each treatment group are shown in Figure 8A, showing less structural damage and increased proteoglycan staining (red) in the encapsulated cell group. This scoring system assigns a number from 0–6 based on how much of the articular cartilage surface is intact and healthy. The higher the score, the more damage to the joint. All articular cartilage surfaces were scored by a blinded technician, and the scores were summed for each joint. Results are displayed as the average summed OARSI score per group. As shown in Figure 8B, encapsulated cells resulted in a significantly lower OARSI score compared to the sham group and hydrogel only.

## 4. Discussion

Over the last few decades, MSCs have garnered great interest for their potential to aid in the treatment of OA. However, their success has been limited by insufficient in vivo delivery resulting in inconsistent outcomes [70]. Thus, it is believed that the development of an appropriate delivery system capable of addressing these issues will enable a more viable product with a longer retention time in the joint, thus contributing to successful clinical translation. The work described in this report demonstrated the potential of HA-based hydrogel microspheres for biomaterial-assisted MSC transplantation.

It was originally hypothesized that the mechanism of action of MSC therapy for OA relied on the cell’s ability to differentiate into chondrocytes. However, it is now appreciated that a myriad of factors contribute to their efficacy—with a focus on immunomodulation via the secretion of paracrine factors and cell-to-cell contact [12]. HA-based hydrogel microspheres are ideal for this application because their use can facilitate each of these therapeutic mechanisms. The MSC secretome plays a major role in protecting injured tissues and promoting tissue repair [9]. In the context of OA, six features of the MSC secretome are important for both inhibiting the progression of OA and promoting tissue repair. These features include the secretion of anti-apoptotic, anti-fibrotic, anti-catabolic, pro-chondrogenic, pro-angiogenic, and immunomodulatory factors [9]. We have previously reported on the diffusion characteristics of HA-based microspheres, showing that molecules up to 70 kDa diffuse easily throughout an HA microsphere network [36]. While the microspheres used in this study were composed of a slightly different formulation, the higher MSD of the microspheres used in this study (AHA 1) compared to the previously published formulation (AHA 2) indicated an even greater ability of soluble molecules to diffuse through the microspheres (Figure 2B). Thus, we hypothesize that the microspheres used in this study allowed for molecules under 70 kDa released from encapsulated MSCs to diffuse out of the hydrogel network.

Another hypothesized therapeutic mechanism of MSCs for OA is cell-to-cell contact [39]. For example, MSCs possess the inhibitory molecule programmed death (PD-1) on their surface and can interact with PD-L1 on the surface of B-cells present in an OA joint [39]. This interaction inhibits the proliferation of B-cells and hinders inflammation [39]. Delivery via HA-based hydrogel microspheres can also facilitate transplanted cell-to-host contact, as hydrogels composed of HA are susceptible to enzymatic degradation via hyaluronidase, a native enzyme in both experimental rodent models and human patients [33,71]. This phenomenon is clearly illustrated in Figure 4, where in accelerated enzyme-catalyzed degradation conditions, cell-laden microspheres were able to release encapsulated bMSCs that then adhered to the bottom of the tissue culture flask. Further, it is clear that microsphere fragments were present in the joint for up to 6 weeks, which is significantly longer than the retention time of MSCs delivered without a hydrogel reported in the literature [17,72]. These data are notable because the transient presence of MSCs in the IA space when they are injected in solution is posited to contribute to the limited success seen with these types of transplants [72]. From these data, alongside the increased expression of CD44 (Figure 3B), which is a cell surface receptor that allows MSCs to bind to HA, we hypothesize that delivery in the HA microspheres allows for an extended presence of cells in the joint space, where local, controlled cell release from the microspheres is achieved by hyaluronidase-mediated enzymatic degradation.

Hydrogel microrheology must also be taken into account when considering encapsulated stem cell behavior. Figure 2A illustrates the viscoelastic behavior of the HA-based microspheres. Viscoelasticity has an impact on viability, proliferation, gene expression, and various other cellular processes [73,74,75]. The ability of encapsulated cells to interact with the hydrogel they reside in represents an important consideration for designing cell delivery vehicles [18]. Hydrogels closely mimic the extracellular matrix (ECM) of tissues, in which cells are constantly probing and responding [76]. Thus, the mechanical properties of the hydrogel and resulting cell mechanotransduction will affect cellular behavior [77]. In the context of MSC delivery into osteoarthritic joints, where cell death upon transplantation is a major concern, the viscoelasticity of the hydrogel was essential to the design of the HA microspheres. MSCs are susceptible to anoikis when cell-ECM attachment regions are lost [18]. By providing a structure for encapsulated bMSCs to attach to and interact with, death via anoikis was avoided.

Figure 3C shows that encapsulated MSCs had increased expression of the chondrogenic markers ACAN and SOX9 after 1 week in culture. Several groups have reported changes in cell gene expression when encapsulated in a 3-dimensional network. Guided differentiation in hydrogels is most often carried out using specialized differentiation media [59]. However, control groups cultured without differentiation media can also show upregulated expression of key markers. Burdick et al. reported increased expression of chondrogenic differentiation markers in MSCs cultured in HA-based hydrogels and posited that the scaffold alone could promote chondrogenesis [59]. The mechanism of such a phenomenon can likely be attributed to the interaction between HA and the MSC cell surface receptor CD44. The interaction of CD44 and HA is vital for cartilage homeostasis and allows MSCs to bind to proteoglycans found in cartilage. The upregulation of ACAN and SOX9 (Figure 3B) is hypothesized to be a result of the presence of HA.

Further, the rheological microenvironment experienced by MSCs has been shown to directly impact their gene expression and differentiation lineage. By looking at the G′ and G″ at 0.16 Hz, the frequency at which cells are thought to probe their matrix [55], further insight into what cells are experiencing can be gleaned. As shown in Figure 2A, at 0.16 Hz, the G′ and G″ are 0.1 and 0.15 Pa, respectively, which indicates that cells are experiencing both elasticity and viscosity in the microenvironment. This is of note due to the implication that viscosity has on the chondrogenic differentiation of MSCs. A recent study by Lee et al. demonstrated that MSCs cultured in high-viscosity hydrogels increased the production of cartilaginous matrix components, such as aggrecan [78]. Of note, Figure 2A shows the crossover frequency, defined as the frequency at which G′ equals G″. This value is often measured for IA supplements and compared to the crossover frequency of healthy synovial fluid (0.4 Hz) [41]. The microspheres used in this study were formulated to have a crossover frequency similar to that of healthy synovial fluid in an effort to recapitulate the rheology of the joint space. As shown in Figure 2A, the crossover frequency of the microspheres used in this study was 0.3 Hz. Based on recent reports concluding that both the rheological and biological features of healthy synovial fluid must be recapitulated to achieve successful chondrogenesis [79], we hypothesize that both the microsphere composition (HA) and its similar rheological properties as healthy synovial fluid play a role in the upregulation of chondrogenic markers. There have been reports of limited or unstable differentiation processes when MSCs are exposed to the hostile components seen in osteoarthritic synovial fluid [80]. Though the therapeutic mechanism of MSCs in treating OA does not rely on the differentiation of transplanted cells into functional cartilage cell types, unmonitored differentiation could pose a risk and should certainly be assayed [37]. However, it is important to note that the cells did not lose their stemness factors during the testing period. They still had high expression levels of CD90 and CD29, although lower than unencapsulated cells.

Another consideration that must be made for the use of hydrogel microspheres as stem cell delivery vehicles is the dosing of both the material and the MSCs. The IA space of the animal model used in this study allows for up to 100 µL of solution or fluid to be injected. Thus, roughly 100 µL of cell-laden hydrogel microspheres containing approximately 2 × 10^6^ bMSCs were injected into the afflicted joints. Further, the volume of implanted hydrogel must be carefully considered to avoid pain. HA-based lubricants are already used clinically for viscosupplementation, but often pain and swelling are early side effects of such procedures. The pain score data in Figure 5 illustrates significantly lower pain scores in the encapsulated cell group one week and three weeks after the transplant. Interestingly, significantly decreased pain is not reflected in the hydrogel-only and cells-only groups at these time points which contained the same volume of microspheres or cells, respectively, as the encapsulated cell group. Pain relief following an MSC transplant into the IA space has been well documented [81], but oftentimes the data are variable [15]. Interestingly, after 3 weeks, there were no significant differences in pain between experimental groups. It is hypothesized that this is a limitation of the pain assay, as animals grow more accustomed to being handled over time and are therefore less sensitive to manual stimulation of the knee joint [82]. However, the important conclusion of the study is that there was no statistical increase in pain levels in the rats when injecting fully crosslinked microspheres.

Histological analysis of the afflicted joints provided further insight into the therapeutic functionality of cell-laden microspheres as MSC delivery vehicles. First, there was dramatically more subchondral bone in all the treatment groups compared to healthy knees that did not undergo OA induction. Thus, while all groups showed excessive subchondral bone, there were no significant differences between the experimental groups (Figure 7). In sharp contrast, treatment with encapsulated cells resulted in significantly less OA damage on the articular cartilage surface (Figure 8). The relationship between subchondral bone remodeling and articular cartilage erosion has been up for debate, but many recent reports suggest that subchondral bone remodeling is evident in the early stages of OA and may even precede the formation of articular cartilage lesions [68,83,84]. While the sequence of events depends on the model used, it has recently been shown that cartilage destruction in post-traumatic injury OA models is secondary to pathological changes in the subchondral bone [83]. With significantly less damage to the articular cartilage surface in the encapsulated cells group (Figure 7B), we hypothesize that the cell-laden microspheres used in this study delayed the onset of cartilage erosion following pathological subchondral bone remodeling.

## 5. Conclusions

Taken together, the data demonstrate that HA-based hydrogel microspheres are beneficial as a cell delivery vehicle when treating OA by supporting the high viability of encapsulated bMSCs, maintaining the stemness of MSCs while depressing the expression of osteogenic biomarkers, increasing the retention time of the hydrogel in rat joints, and improving joint integrity in an OA rat model. As the retention time of stem cell injections into weight-bearing joints has been a challenge for the field, the benefits offered by fully crosslinked, injectable microspheres may be useful in improving patient outcomes.

## Figures and Tables

**Figure 1 bioengineering-10-01315-f001:**
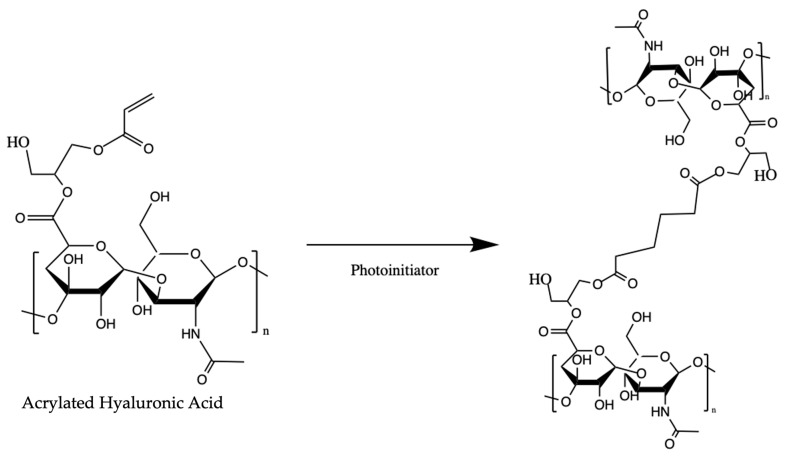
Photopolymerization of AHA illustrating the resulting crosslinks between polymer chains.

**Figure 2 bioengineering-10-01315-f002:**
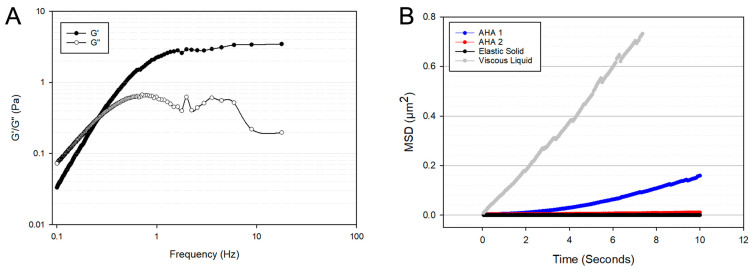
MPT Microrheology of HA-based hydrogel microspheres. (**A**) The G′ and G″ of the HA microspheres utilized in this study. The point where the G′ and G″ overlap is called the crossover frequency and the hydrogel formulation was designed to have a crossover frequency equivalent to healthy joint synovial fluid. (**B**) The permeability profile of the HA microspheres used in this study (AHA 1, blue) was compared to a previously published formulation (AHA 2, red) and are shown next to the MSD of model materials, including that of a viscous liquid (gray) and an elastic solid (black). The higher MSD value is indicative of a more permeable hydrogel network.

**Figure 3 bioengineering-10-01315-f003:**
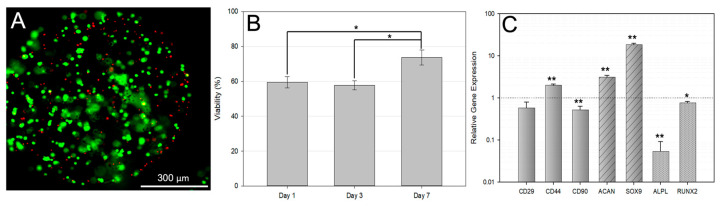
Effect of HA Hydrogel Microspheres on Encapsulated Cells. (**A**) An image of a single microsphere stained for live and dead cells 7 days after encapsulation is shown. The green cells are living, while the red indicates dead cells. The scale bar is 300 microns. (**B**) The viability of encapsulated bMSCs over 7 days, as determined by Live/Dead staining, was shown to improve over time. (**C**) qPCR analysis of encapsulated bMSCs after 7 days in culture determined the impact of HA microspheres on bMSC gene expression. Values were normalized to unencapsulated bMSCs after 7 days in culture and exposure to the same enzymatic digestion as encapsulated bMSCs. The dotted line indicates no change in gene expression compared to the unencapsulated cells. Larger numbers indicate a higher gene expression, while lower values indicate less expression. * *p* < 0.05, ** *p* ≤ 0.001.

**Figure 4 bioengineering-10-01315-f004:**
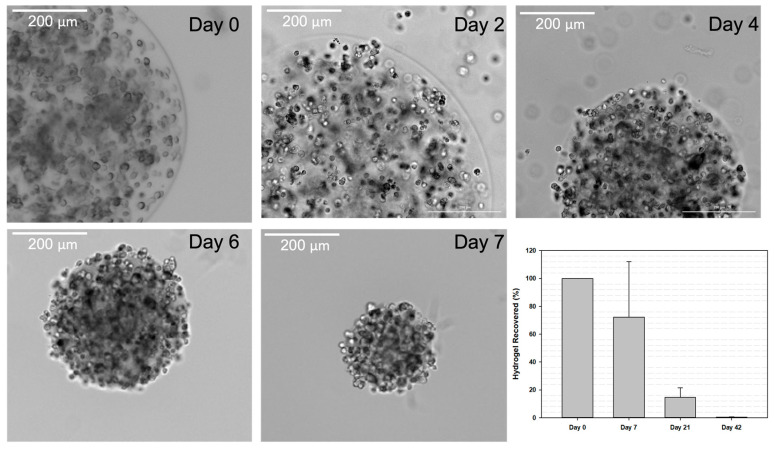
In Vitro and In Vivo Degradation of HA Microspheres. HA microspheres were stored in 50 U/mL hyaluronidase to simulate an accelerated enzyme-catalyzed degradation. The images provide examples of bMSC-laden microspheres prior to the addition of hyaluronidase (day 0) and at 2, 4, 6, and 7 days later. The graph summarizes the in vivo retention of HA microspheres in rat knee joints. Labeled hydrogel material was recovered from the joints 1, 3, and 6 weeks after IA injection. Data were normalized to the amount of hydrogel present in the initial injection (Day 0). Scale bars = 200 microns.

**Figure 5 bioengineering-10-01315-f005:**
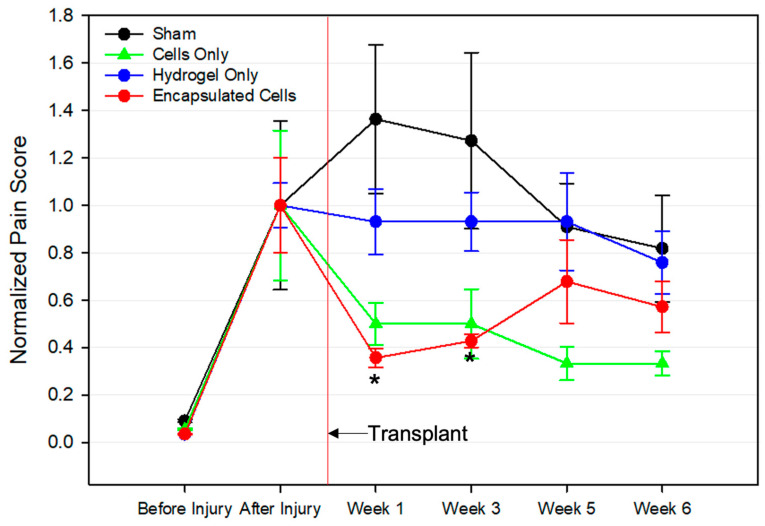
Pain Scores Associated with Treatment Groups. Different treatments were injected into the knees of rats 4 weeks after a traumatic injury model for OA was induced. The first pain assessment was made before OA induction (before injury). Just prior to the injection of the treatments, another pain assessment was recorded (after injury). The red vertical line indicates when the treatments were injected. Pain scores were collected in the following weeks until the completion of the study. Only the group that received encapsulated cells had a statistically lower pain score than the pre-treatment group, and that was only on weeks 1–3. Data are normalized to the pre-transplant value. * *p* < 0.05.

**Figure 6 bioengineering-10-01315-f006:**
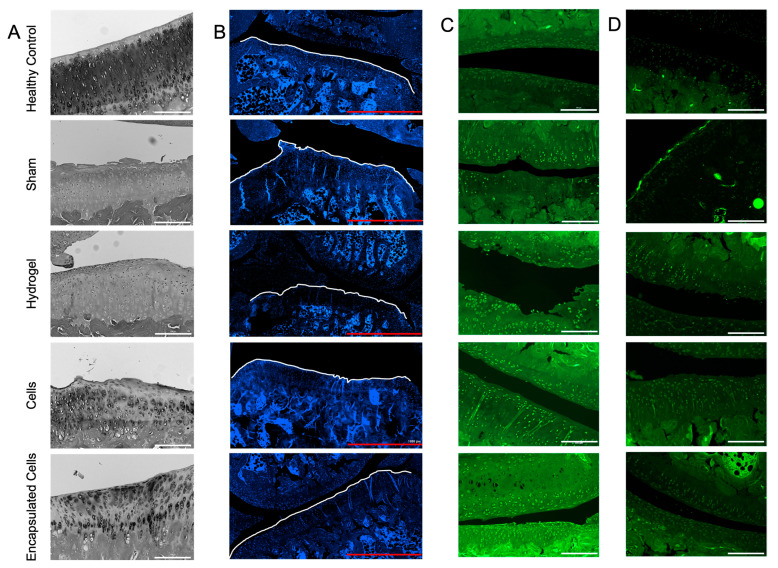
Joint histology in OA rats. (**A**) The bright field images illustrated the uneven surface of the bone for the groups except for the healthy controls and the rats receiving encapsulated cells. (**B**) Dapi-staining of the joint sections was used to determine cell density at the surface. The images also illustrate the rough bony surfaces, outlined by the white line. (**C**) Collagen II staining illustrated a punctate staining pattern in the injured joints but not in the joints of healthy, non-injured animals. (**D**) ADAMTS5 staining produced no obvious difference between groups except for the enhanced surface staining in the sham group. White scale bars = 200 μm, red scale bars = 1000 μm.

**Figure 7 bioengineering-10-01315-f007:**
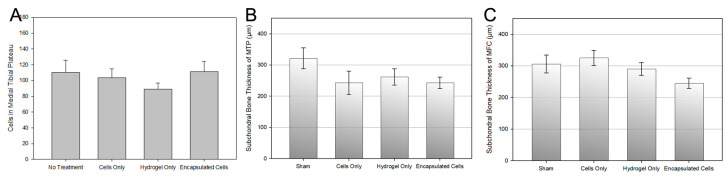
Quantification of Histological Differences Between Groups. (**A**) The number of cells at the bone surface were counted using Dapi-stained sections. There was no statistical difference between the groups. (**B**) The thickness of the subchondral bone in the medial tibial plateau and (**C**) the medial femoral condyle are shown. Treatment with encapsulated cells resulted in the least amount of subchondral bone thickness, although there were no significant differences between experimental groups as determined by a one-way ANOVA.

**Figure 8 bioengineering-10-01315-f008:**
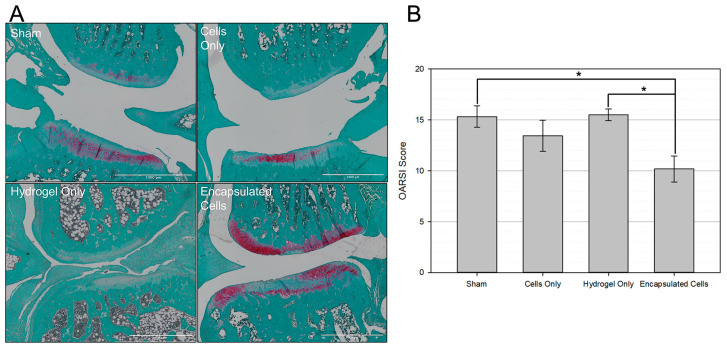
Encapsulated Cells Associated with Improved Articular Cartilage Surface. (**A**) Representative images of the Medial Femoral Condyle (Top) and Medial Tibial Plateau (bottom) for each experimental group are shown. Sections were stained with Safranin-O and Fast Green which stain proteoglycans and collagen, respectively. (**B**) The OARSI histopathological score for OA damage was used on each experimental group to assess the degree of osteoarthritic damage to the articular cartilage surface. Only the joints receiving encapsulated bMSCs showed a statistical improvement in articular cartilage scoring. * *p* < 0.05.

**Table 1 bioengineering-10-01315-t001:** Assay identification numbers used in the study.

	Assay ID
GUSB	Rn00566655_m1
CD34	Rn03416140_m1
CD45	Rn00709901_m1
CD90	Rn00562048_ml
CD29	Rn00562048_m1
CD44	Rn00681157_m1
ACAN	Rn00573424_m1
SOX9	Rn01751069_mH
Col2a1	Rn01637087_m1
ALPL	Rn01516028_m1
RUNX2	Rn01512298_m1

**Table 2 bioengineering-10-01315-t002:** Physical properties of AHA microspheres.

	AHA Microspheres
Polymer mass fraction in gel precursor	5% AHA 200 kDa 2% 1 kDa PEGDA
Precursor viscosity	116 cST
Sphere size range	627–1347 µm
Polymer mass fraction in final spheres	0.9%
Mass swelling ratio “Q”	108 ± 3
Average final diameter	1036 ± 180 µm

**Table 3 bioengineering-10-01315-t003:** Assay identification numbers and C_t_ values for unencapsulated and encapsulated MSCs.

	Unencapsulated MSCs	Encapsulated MSCs
GUSB	27.75 ± 0.64	28.12 ± 0.64
CD34	Not detected	Not detected
CD45	Not detected	Not detected
CD90	27.13 ± 0.56	28.46 ± 0.50
CD29	26.19 ± 2.16	27.41 ± 2.31
CD44	26.45 ± 0.54	25.82 ± 0.61
ACAN	23.42 ± 0.78	22.14 ± 0.90
SOX9	30.51 ± 1.11	26.67 ± 1.24
Col2a1	Not detected	Not detected
ALPL	28.97 ± 0.75	33.99 ± 1.61
RUNX2	29.68 ± 1.47	30.45 ± 1.58

## Data Availability

All data generated and/or analyzed during this study are available from the corresponding author upon reasonable request.

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
