# Peer review of "Controlled-Release Hydrogel Microspheres to Deliver Multipotent Stem Cells for Treatment of Knee Osteoarthritis"

_bioengineering, 2023, doi:10.3390/bioengineering10111315_

Round 1
Reviewer 1 Report
Comments and Suggestions for Authors
Dear authors,
The authors present the controlled-release hydrogel microspheres to deliver Multipotent Stem Cells for treatment of knee osteoarthritis. The manuscript is scientifically sound. However, a minor revision is required before accepting it for publication.
1. The authors are requested to provide a scheme showing the synthesis and probable crosslinking mechanism of the prepared hydrogel.
2. In the Figure 3, scale bars are not visible.
3. Minor English editing is required.
Thanking you,
Comments on the Quality of English Language
Minor English editing is required.
Author Response
- The authors are requested to provide a scheme showing the synthesis and probable crosslinking mechanism of the prepared hydrogel.
Thank you for the suggestion. The manuscript has been updated to include a scheme that illustrates the crosslinks between polymer chains that form during radical photopolymerization of the AHA hydrogel microspheres. It is the new Figure 1.
- In the Figure 3, scale bars are not visible.
Figure 3 has been revised to include scale bars.
- Minor English editing is required.
The authors have re-read the manuscript and made changes to grammatical errors.
Reviewer 2 Report
Comments and Suggestions for Authors
This study by Hamilton et al., demonstrate the potential of using HA-based hydrogel microspheres to enhance the therapeutic efficacy of MSCs in treating OA.
The study is exciting, addressing some important aspects in the area of sustained delivery of cells in the knee joint.
Introduction:
Authors should include an outline/summary of the study at the end of the Introduction in a few lines.
Methods:
Line #186 - 100 million cells? Is that right?
Authors should include primer sequences for the targets.
Results:
It will be interesting to know how many (Or what %) cells were left at the time of harvest.
Authors should consider cell viability (in vitro) assay for the duration of 4 week (the in vivo assay time point)
Discussion/Conclusion:
Good
References:
Ref. #3 authors should cite original reference for stats.
Author Response
- Authors should include an outline/summary of the study at the end of the Introduction in a few lines.
Thank you for this suggestion. The introduction has been updated to summarize the studies (lines 133-140)
- Line #186 - 100 million cells? Is that right?
Thank you for this correction. It was actually 80 million cells per mL of precursor. This value was chosen based on the swelling behavior of these microspheres, as well as the required dosing/volume needed for in vivo transplants. Prior to cell encapsulation, it was determined that for every 1 mL of hydrogel precursor, 4 mL’s of hydrogel microbeads were obtained. So at 80 million cells/mL of precursor the final cell seeding density in 1 mL of hydrated hydrogel microspheres was 20 million cells/mL. This was chosen for this application because the goal was to inject 2 million cells total into each rat joint, in a volume of 0.1 mL’s. Thus, at 20 million cells/mL of hydrated beads, the final cell number in 0.1 mL’s was 2 million cells, which was ideal for injection into the small volume of a rodent IA space. We have updated the methods section to include the rationale for this cell seeding density (lines 246 – 252).
- Authors should include primer sequences for the targets.
Thermo Fisher Scientific does not make public the primer sequences, so we’ve added the vendor’s Assay ID numbers to Table 2 to ensure that the methods are clear and reproducible.
- It will be interesting to know how many (Or what %) cells were left at the time of harvest.
The authors agree with the reviewer’s comment. This manuscript is a proof-of-concept study for use of our hydrogel microspheres as a delivery vehicle. We have current, ongoing work that addresses this comment using human MSCs. The goal of our ongoing work is to quantify how long cells are retained in the joint space post-transplant, as well as to quantify how many cells are left in the microspheres at the end of the study.
- Authors should consider cell viability (in vitro) assay for the duration of 4 week (the in vivo assay time point)
The authors agree with this comment. For this study, the goal was to ensure that encapsulated cells were viable at the time of transplant. Ongoing work for another manuscript using human rather than rat MSCs is being carried out to assess viability and gene expression throughout the entire lifetime of a simulated transplant to address questions such as this.
- #3 authors should cite original reference for stats.
An original paper has now been referenced.
Reviewer 3 Report
Comments and Suggestions for Authors
The current work by Hamilton et describes the use of acrylated hyaluronic acid (HA-MA) and polyethylene glycol (PEGDA) based-hydrogel micro-carriers as mesenchymal stem cell delivery vehicles for treating osteoarthritis. Stehno-Bittel’s group have previously explored the fabrication strategy and the materials used here (HA-MA and PEGDA) for stem cell delivery: Tissue Eng., Part A 2021, 27, 153-164; ACS Biomaterials Science & Engineering 2021, 7, 3754-3763.
Here are few suggestions which the authors should consider for improving the manuscript:
1. In introduction/ abstract, the authors must mention that this study is a follow-up study taking inspiration from their earlier published works. It would also be beneficial if the introduction can bring the relevance/ significance of the current work in comparison to their previous works (the novelty aspect?).
2. For modification of hyaluronic acid, the methods followed in Tissue Eng., Part A 2021, 27, 153-164 and ACS Biomaterials Science & Engineering 2021, are different but they attain the same degree of modification – degree of acrylation 50-70 %; in the current work they have followed ACS Biomaterials Science & Engineering 2021 protocol. Is there a reason why this was chosen? This could be explained in discussion section.
3. HA-based hydrogels are used for OA applications. Injectable hydrogels which are cured by photo-initiators are present in literature. How would the micro-carriers prepared with such complexity be any different from a more facile delivery of just injectable hydrogel?
(See references: Kowalski, et al. "Cartilage‐penetrating hyaluronic acid hydrogel preserves tissue content and reduces chondrocyte catabolism." Journal of Tissue Engineering and Regenerative Medicine 16.12 (2022): 1138-1148; Patel, Jay M., et al. "Stabilization of damaged articular cartilage with hydrogel‐mediated reinforcement and sealing." Advanced healthcare materials 10.10 (2021): 2100315.)
4. The reviewer understands that bulk hydrogel (especially alignate-based) could limit cell survival, but HA- based hydrogels can be remodelled in the OA lesion space, and wouldn’t the diffusion coefficient favour hypoxic conditions leaning towards chondrogenic differentiation? Moreover, retention of microgel micro-carriers versus hydrogels was not discussed. The authors need to focus on more recently published works and reframe their introduction/ discussion to bring in this narrative.
5. From micro-rheology (Figure 1), the yield stress, linear viscoelastic region (LVER) and storage modulus values should be provided to understand its mechanical properties and whether it would be relevant for articular cartilage repair.
6. Figure 2A, no scale bar presented. More images with higher magnification should be presented.
7. (Table -2) qPCR studies what was the sample size? No standard error or standard deviation presented. It is confusing because, Figure 2C (which is not labelled in the panel), has data which is different from the table. There is no clarity in presentation. Expression of runx2, indicates mineralization which is not ideal for cartilage regeneration. Authors need to address this.
8. Figure 3 scale bars are missing. Images must be provided in same magnification for better clarity and understanding.
9. Figure 4, the hydrogel only implies only micro carrier group in legend/ label? Can this then be labelled as non-encapsulated group from this figure onwards for better clarity?
Figure 5, scale bars are missing. Images seem to have lot of autofluorescence. How was the quantification carried out in Figure 6, to avoid this autofluorescence?
Table-1 line 216, PCR primers missing/ wrongly labelled.
Author Response
Reviewer 3:
- In introduction/ abstract, the authors must mention that this study is a follow-up study taking inspiration from their earlier published works. It would also be beneficial if the introduction can bring the relevance/ significance of the current work in comparison to their previous works (the novelty aspect?).
Thank you for this suggestion. The introduction has been updated to illustrate the significance of this work compared to our earlier work. Two paragraphs have been added to the introduction (Lines 104-128).
- For modification of hyaluronic acid, the methods followed in Tissue Eng., Part A 2021, 27, 153-164 and ACS Biomaterials Science & Engineering 2021, are different but they attain the same degree of modification – degree of acrylation 50-70 %; in the current work they have followed ACS Biomaterials Science & Engineering 2021 protocol. Is there a reason why this was chosen? This could be explained in discussion section.
Thank you for this question. The methods followed in Tissue Eng., Part A 2021, 27, 153-164 were to functionalize HA with a methacrylate group. The methods followed in this manuscript and in ACS Biomaterials Science & Engineering 2021 were for functionalization of HA with an acrylate group, which is one methyl group short of methacrylate. Thus, the methods used for modification were different because they were attaching different functional groups. Additionally, we have now added a scheme showing this structure and crosslinking to illustrate this point- new Figure 1.
- HA-based hydrogels are used for OA applications. Injectable hydrogels which are cured by photo-initiators are present in literature. How would the micro-carriers prepared with such complexity be any different from a more facile delivery of just injectable hydrogel?
This is an excellent question. The rationale for using HA microspheres is that they have the potential to facilitate higher levels of encapsulated cell viability by presenting less of a diffusion barrier for nutrient/waste exchange compared to bulk hydrogels. Additionally, the therapeutic mechanism of MSCs in treating OA relies, in part, on the diffusion of paracrine & immunomodulatory factors that have been released by encapsulated MSCs. Thus, those soluble factors need to be able to diffuse out of the microsphere. We recognize your later comment about diffusion in the context of cartilage regeneration and have answered that below. We have added this relevant context to the introduction (lines 112-118)
(See references: Kowalski, et al. "Cartilage‐penetrating hyaluronic acid hydrogel preserves tissue content and reduces chondrocyte catabolism." Journal of Tissue Engineering and Regenerative Medicine 16.12 (2022): 1138-1148; Patel, Jay M., et al. "Stabilization of damaged articular cartilage with hydrogel‐mediated reinforcement and sealing." Advanced healthcare materials 10.10 (2021): 2100315.)
- The reviewer understands that bulk hydrogel (especially alignate-based) could limit cell survival, but HA- based hydrogels can be remodeled in the OA lesion space, and wouldn’t the diffusion coefficient favour hypoxic conditions leaning towards chondrogenic differentiation? Moreover, retention of microgel micro-carriers versus hydrogels was not discussed. The authors need to focus on more recently published works and reframe their introduction/ discussion to bring in this narrative.
The authors recognize the value that bulk hydrogels have in efforts on cartilage remodeling and chondrogenic differentiation of encapsulated MSCs. There is an important distinction between stem cell therapy and tissue engineering, which are two completely different modes of therapeutic action. For stem cell therapy, MSCs function therapeutically by releasing soluble factors that interact with the host tissue, and this function actually relies on MSCs maintaining their stemness, not on their differentiation into functional chondrocytes, which is in contrast to the goal of tissue engineering seeking to differentiate the cells into new cartilage.
The study described in this manuscript aims to address limitations within stem cell therapy, not tissue engineering, which we pointed out in the original manuscript (lines 60-64). Thus, while it may be beneficial for the diffusion coefficient to favor hypoxic conditions towards achieving chondrogenic differentiation within the field of tissue engineering, that is not ideal for this application where the goal is to maintain MSC identity and potency, and for soluble factors released by the MSCs to diffuse out of the hydrogel microsphere. We have added this addition text to the introduction to further explain the differences in the two approaches (lines 110-118).
- From micro-rheology (Figure 1), the yield stress, linear viscoelastic region (LVER) and storage modulus values should be provided to understand its mechanical properties and whether it would be relevant for articular cartilage repair.
The microrheology work done in this manuscript is based on frequency sweeps. Contrastingly, yield stress, LVER, and elastic modulus are obtained from stress-strain curves at one strain rate – not from a frequency sweep. It is not possible for traditional viscoelastic instruments, such as a Dynamic Mechanical Analysis or bulk rheometer, to be used on a single hydrogel microsphere that we were able to accomplish with microrheology.
The output from frequency sweeps, which is shown in Figure 2A, is the storage (G’) and loss (G”) moduli, which give one insight into the relative elastic and viscous contributions to a materials viscoelasticity. In the context of this study, the microrheology is the only method available to answer the question of crossover frequency (defined as the frequency at which G’ becomes greater than G”) as this is value is particularly relevant for materials in the articular space and has been well characterized for HA-based viscosupplements, as we explained in the discussion of the original manuscript (currently lines 804-818) and described by reference 40.
- Figure 2A, no scale bar presented. More images with higher magnification should be presented.
We have edited Figure 2A (now Figure 3A) to a higher magnification image and included the scale bar.
- (Table -2) qPCR studies what was the sample size? No standard error or standard deviation presented. It is confusing because, Figure 2C (which is not labelled in the panel), has data which is different from the table. There is no clarity in presentation. Expression of runx2, indicates mineralization which is not ideal for cartilage regeneration. Authors need to address this.
Thank for this feedback as we failed to include the repetition number in the methods section. We have corrected that with lines 294-295 of the methods section.
The qPCR study was performed on 3 independent batches, each in triplicate. In addition, the standard deviation for the Ct values in Table 2 has been added and the Figure 3C is now labeled. The data presented in Figure 3C area the RQ values, or relative gene expression in encapsulated cells compared to unencapsulated cell as described in the original methods section (lines 294-297). The RQ value was calculated using the common method based on . In Figure 3C, the expression of RUNX2 was statistically lower in the encapsulated cells compared to unencapsulated, as was described in the figure caption in the original version of the manuscript.
- Figure 3 scale bars are missing. Images must be provided in same magnification for better clarity and understanding.
We have edited Figure 3 (now Figure 4) to include scale bars.
- Figure 4, the hydrogel only implies only micro carrier group in legend/ label? Can this then be labelled as non-encapsulated group from this figure onwards for better clarity?
Thank you for this clarifying question. “Hydrogel Only” means that it was the hydrogel microspheres with no encapsulated cells. If we were to label this as “non-encapsulated” group that would imply it was an injection of cells with no encapsulant, and that would add confusion for the reader.
- Figure 5, scale bars are missing. Images seem to have lot of autofluorescence. How was the quantification carried out in Figure 6, to avoid this autofluorescence?
The scale bars for Figure 5 (now Figure 6) have been adjusted for clarity. The quantification of cell density shown in Figure 6 was based on the DAPI stained images with a focus on the articular cartilage surface, where there was less autofluorescence than in the subchondral bone and surrounding tissues. The images were taken with the same exposure time, intensity, and gain, then uploaded into ImageJ where the same brightness/contrast settings were applied prior to cell counting to ensure consistency between all images. This description has been added to the methods section (lines 394-400).
The other images were provided as examples only. There was no analysis of the images conducted because, by eye, there were no obvious differences between groups as described in the original manuscript. The image collection was purposefully designed to catch some background staining so that the location of the surface of the joint could be determined. Without the enhanced tissue staining it was impossible to determine the location of the joint surface.
- Table-1 line 216, PCR primers missing/ wrongly labelled.
Thank you for this comment. Thermo Fisher Scientific does not make the primer sequences publicly available, so we’ve added the Thermo Fisher Assay ID numbers to Table 2 to ensure that the methods are clear.
Round 2
Reviewer 2 Report
Comments and Suggestions for Authors
The authors have made necessary revisions and answered the queries.
Reviewer 3 Report
Comments and Suggestions for Authors
The authors have answered the queries satisfactorily. The manuscript is deemed fit for publication pending editorial acceptance.